# Applying heat and humidity using stove boiled water for decontamination of N95 respirators in low resource settings

**Siddharth Doshi[1], Samhita P. Banavar[2], Eliott Flaum[2,3], Surendra Kulkarni[4],
Ulhas Vaidya[4], Shailabh Kumar[2], Tyler Chen [2], Arnab Bhattacharya[4], Manu Prakash [2]***

**1** Department of Materials Science and Engineering, Stanford University, Stanford, California, United States of America, **2** Department of Bioengineering, Stanford University, Stanford, California, United States of America, **3** Graduate Program in Biophysics, Stanford University, Stanford, California, United States of America, **4** Tata Institute of Fundamental Research, Mumbai, India

* manup@stanford.edu

**Data Availability Statement:** Data are available at: https://doi.org/10.5281/zenodo.4014348.

**Funding:** SD is supported by an Office of Technology and Licensing Stanford Graduate

## Abstract

Global shortages of N95 respirators have led to an urgent need of N95 decontamination and reuse methods that are scientifically validated and available world-wide. Although several large scale decontamination methods have been proposed (hydrogen peroxide vapor, UV-C); many of them are not applicable in remote and low-resource settings. Heat with humidity has been demonstrated as a promising decontamination approach, but care must be taken when implementing this method at a grassroots level. Here we present a simple, scalable method to provide controlled humidity and temperature for individual N95 respirators which is easily applicable in low-resource settings. N95 respirators were subjected to moist heat (>50% relative humidity, 65–80˚C temperature) for over 30 minutes by placing them in a sealed container immersed in water that had been brought to a rolling boil and removed from heat, and then allowing the containers to sit for over 45 minutes. Filtration efficiency of 0.3–4.99 μm incense particles remained above 97% after 5 treatment cycles across all particle size sub-ranges. This method was then repeated at a higher ambient temperature and humidity in Mumbai, using standard utensils commonly found in South Asia. Similar temperature and humidity profiles were achieved with no degradation in filtration efficiencies after 6 cycles. Higher temperatures (>70˚C) and longer treatment times (>40 minutes) were obtained by insulating the outer vessel. We also showed that the same method can be applied for the decontamination of surgical masks. This simple yet reliable method can be performed even without electricity access using any heat source to boil water, from open-flame stoves to solar heating, and provides a low-cost route for N95 decontamination globally applicable in resource-constrained settings.

## Introduction

The ongoing COVID-19 pandemic has caused a worldwide shortage of N95 filtering facepiece respirators (FFRs). Many health facilities are rationing N95 FFRs and reusing them with

Fellowship. The Gordon and Betty Moore Foundation and Schmidt Foundation also supported MP in COVID related work. The National Science Foundation also supported this project through a grant given to MP (DBI-1548297). The funders had no role in study design, data collection and analysis, decision to publish, or preparation of the manuscript.

**Competing interests:** The authors have declared that no competing interests exist.

various decontamination protocols. Nations with emerging economies and high population densities (e.g. India, Pakistan, Bangladesh, Brazil, Peru, Ecuador), are expected to face significant public health challenges due to these shortages. Therefore, it is important to develop accessible procedures for decontamination and re-use of N95 FFRs without damaging their fit or filtration efficiency.

At a minimum, decontamination requires inactivation of the SARS-CoV-2 virus and maintenance of the fit and filtration efficiency of the N95 respirator. Temperature treatments, which can inactivate viruses through protein denaturation, represent a potential disinfection method [1]. While there is recent data suggesting that SARS-CoV-2 in liquid media can be inactivated by exposure to 70°C for 5 minutes [2], it appears that the virus requires a much longer heat-treatment for inactivation on N95 FFR surfaces. A recent, non-peer-reviewed report indicates that application of 70°C dry heat for 30 min only led to a 1.9-log reduction in viral load of SARS-CoV-2 in an unknown media on N95 fabric [3], which is below the minimum 3-log inactivation level suggested by the FDA for emergency use authorization for decontamination [4]. Additionally, the media used for inoculation may not be representative of human saliva or mucin, which may falsely increase inactivation rates compared to a real-world scenario [5–8]. While direct studies involving the effects of heat and humidity on the SARS-CoV-2 virus on N95 FFRs are limited, increased humidity was found to increase inactivation rates of the H1N1 influenza virus at 65 °C on steel surfaces [9–11]. A non-peer-reviewed report found that increasing relative humidity from 13% to 48% increased inactivation of MS2 and Phi6 bacteriophages deposited with PBS on N95 FFRs by >4-log during a 72°C heat treatment for 30 min [7]. After this 30 min treatment at 72°C and 48% relative humidity, both viruses were inactivated by over 6-log. MS2 belongs to a class of non-enveloped viruses, which are traditionally considered to be more difficult to inactivate than enveloped viruses such as SARS-CoV-2 [12]. Given this emerging data, it is likely, though unproven, that temperatures of 65°C and relative humidities of >50% applied for at least 30 minutes will lead to at least 3-log inactivation of SARS-CoV-2 on an N95 FFR.

While increasing temperature and humidity may improve viral inactivation, it also carries the risk of damaging N95 fit and filtration. Several N95 FFR models have been shown to fail fit tests after greater than one 121°C autoclave treatment cycle, indicating that temperature, humidity, and duration of decontamination must be carefully chosen to strike a balance between viral inactivation and N95 performance [13]. Fortunately, many models of N95 FFR have been shown to undergo at least three cycles of elevated temperature (65–85°C) at greater than 50% relative humidity for 20–30 min while maintaining filtration efficacy and fit [14–16], indicating that conditions of moist heat at 65–80 °C, with 50% relative humidity for 30 minutes may be a promising target for decontamination of SARS-CoV-2 on N95 FFRs. However, reliably achieving these heat and humidity conditions in low resource environments is challenging, particularly those without equipment for heating or stable access to electricity. Equipment typically used for surgical sterilisation (e.g. autoclaves) run at temperatures which, as outlined earlier, may result in a loss of fit and filtration efficiency [14, 15]. There is a need to develop methods applicable to low resource contexts that can achieve 65–80°C temperatures at high humidity (>50%) for over 30 minutes. A method of heating N95 FFRs with moisture to achieve temperatures of 85°C at 65–80% humidity was recently implemented by placing plastic containers containing N95 FFRs and 500 μL of water into an oven [16]. Implementation of a similar method involving equipment available in low resource settings could have wide applicability.

Open flame stoves are widely used for cooking in a range of remote and low resource contexts where access to electricity is limited or intermittent. Using these stoves to achieve heat and humidity conditions could lead to a widely accessible method of decontamination, if

validated properly. Boiling water and cooking utensils are commonly used across India, Bangladesh, Pakistan and many other countries to disinfect surgical equipment in clinical settings lacking traditional autoclaves [17–19]. Regulation of the temperatures inside a cooker or closed cooking vessel could be achieved through the use of high thermal mass elements such as boiling water. Heating of the water to its boiling point at atmospheric pressure provides a reliable way of reaching a set temperature. Due to the high heat capacity of water, heat transfer may occur at slow enough rates to maintain 65–80°C temperatures and >50% humidity for over 30 minutes after the vessel is removed from the stove.

Here we implement a method in which N95 respirators are placed inside containers, which are placed inside larger cooking vessels with water that has been heated to a rolling boil and then removed from heat. We demonstrate that this method can maintain a consistent elevated temperature and humidity as required for decontamination of N95 FFRs using materials available in low-resource settings. This method could be implemented using a wide variety of conditions based on practical considerations and the materials available in a local context. In this work we demonstrate two specific implementations of this protocol. The first, carried out in an air conditioned building (Stanford, CA, USA) uses a 1.65L Pyrex glass container as the FFR holding container, a saturated 5x5cm paper towel with water from a tap as a moisture source to achieve humidity, and a standard 6 quart (5.7L) cooking vessel containing 2L of water as the larger vessel. The second uses a 2L steel container contained within a 5L aluminium vessel and was carried out in a non air-conditioned kitchen environment (Mumbai, MH, India). Specific procedural details are described for each implementation in the sections below.

Additionally, we also demonstrate that a similar decontamination process can be used for surgical masks as well (See Supporting Information S1.4 in S1 File for details).

## Heating protocol

1. A Kimberly Clark N95 respirator was placed in a 1.75 quart (1.65L) Pyrex container (Fig 1a).

2. A small strip of a paper towel (~5x5cm before folding) was folded, doused in water under a tap, squeezed to remove excess water until it no longer dripped passively and then placed in the Pyrex mask container. The container was closed with a tight lid. A BME280 sensor (Sparkfun Inc) was included in the container to log temperature and humidity data. For testing purposes, the respirator was separated into two halves to obtain two filtration data points during the same treatment cycle (Fig 1b). During implementation of this protocol, the N95 mask should be minimally handled.

3. 2L of water was brought to a full rolling boil inside a separate, larger 6 quart (5.7L) vessel, and the **vessel was then removed from heat**. It is important to clarify that a rolling boil, where large bubbles vigorously rise and continually break the surface, was achieved, as opposed to a simmering boil, where small bubbles occasionally break the surface. We turn on logging of testing data from this time point onwards. While this occurred on an electric stove, this method generalises to open flame stoves. The presence of bubbles rapidly breaking the surface was used as a visual marker for boiling, which, neglecting major changes in altitude, reliably indicates water temperatures of close to 100 °C. Water temperatures measured (Kizen Instant Read Meat Thermometer) after moving the vessel off the stove ranged from 93–97 °C.

4. The sealed respirator container was immediately placed in the large vessel containing the boiled water (Fig 1c).

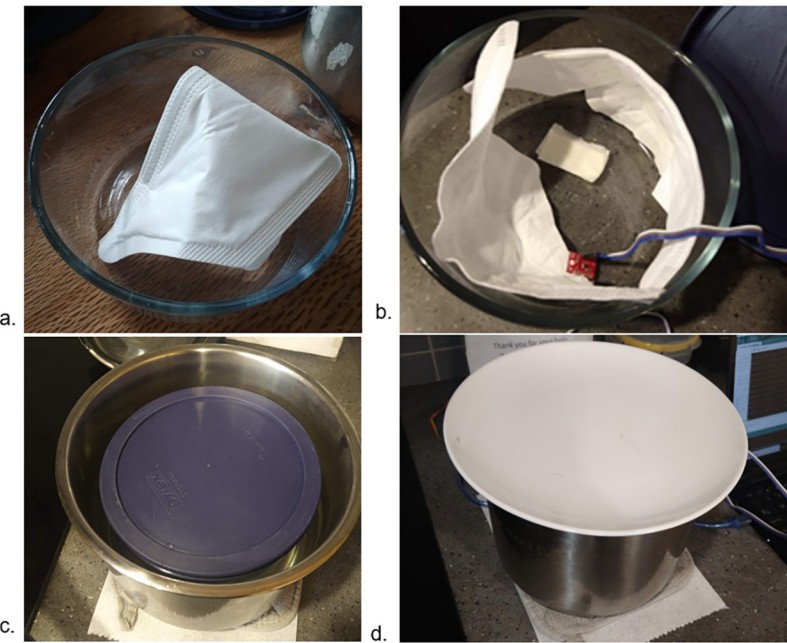

**Fig 1. (a) Kimberly-Clark N95 respirator inside 1.65L (7 cup, 1.75 quart) container, (b) Folded 5x5cm wet paper towel, two halves of the respirator and BME280 sensor inside the container (c) Large 5.7L cooking vessel, removed from stove after heating 2L water to vigorous boil, with a closed 1.65L Pyrex container respirator floating inside, (d)Vessel covered with a plate and allowed to sit for at least 45 minutes while data is logged.**

5. The large vessel was covered with a non-airtight lid and allowed to sit with the closed respirator container inside for at least 45 minutes (Fig 1d).

6. The container was first removed from the large vessel and then opened.

The ambient temperature was ~22˚C and measured ambient humidity was ~37%. Temperature and humidity inside the container were measured with a BME280 sensor (Sparkfun Inc) and logged with an Arduino Uno. They are reported in Fig 2.

The minimum temperature target of 65˚C was typically reached after approximately 4–7 minutes and max temperatures of between 75–80˚C were observed. The initial spike in humidity is due to saturation of air with water vapour at low temperatures, followed by a rapid increase in vapour pressure (and hence decrease in relative humidity) upon heating, followed by a gradual increase in humidity as the water in the sealed container evaporates. The rapid drop in humidity occurs when the container is opened at the end of treatment. The minimum humidity target was reached between 10–13 minutes after testing began and consistently reached a maximum of ~60%. The variance in data is partly caused by slight differences in time taken to move the container into the large vessel after removing from boil, water volumes, average water temperature at the end of the boil, precise amount of liquid soaked into the towel and in data logging start time. These are likely to occur during normal implementation of this method in any low resource setting.

Despite these operating variations, across all trials the humidity and temperature remained simultaneously above the minimum targets (after reaching both the 50% humidity and 65˚C temperature targets) for over 30 minutes. A total treatment time of 45 mins allowed the system to ramp up to the targets and then maintain them for 30 minutes.

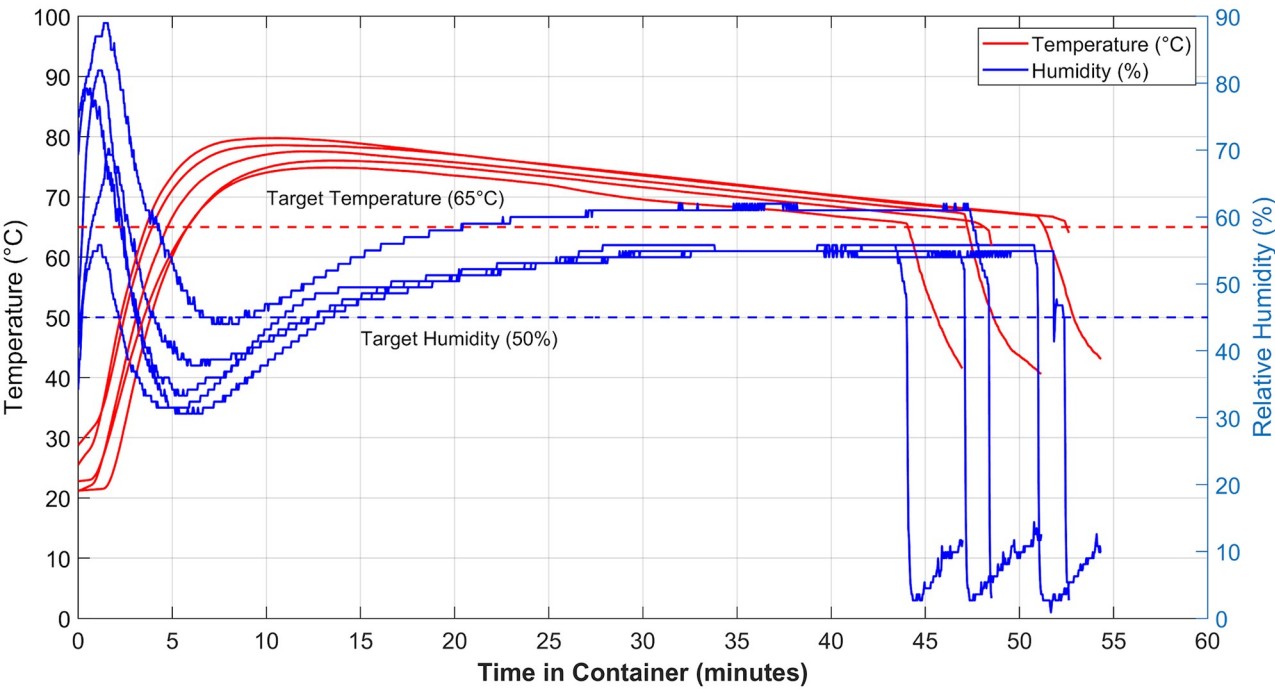

**Fig 2. Temperature and humidity inside the container holding N95 respirators during the 5 treatment cycles after which they underwent filtration testing.** Rapid decrease in humidity in the end is due to opening of the container at the end of the decontamination treatment.

## Filtration efficiency measurements

A simple experimental test rig (Fig 3) and method was used to test the particle filtration efficiency of various materials including N95-grade masks. The setup includes a LightHouse handheld particle counter (Model 3016 IAQ), Incense: Satya Sai Baba Nag Champa

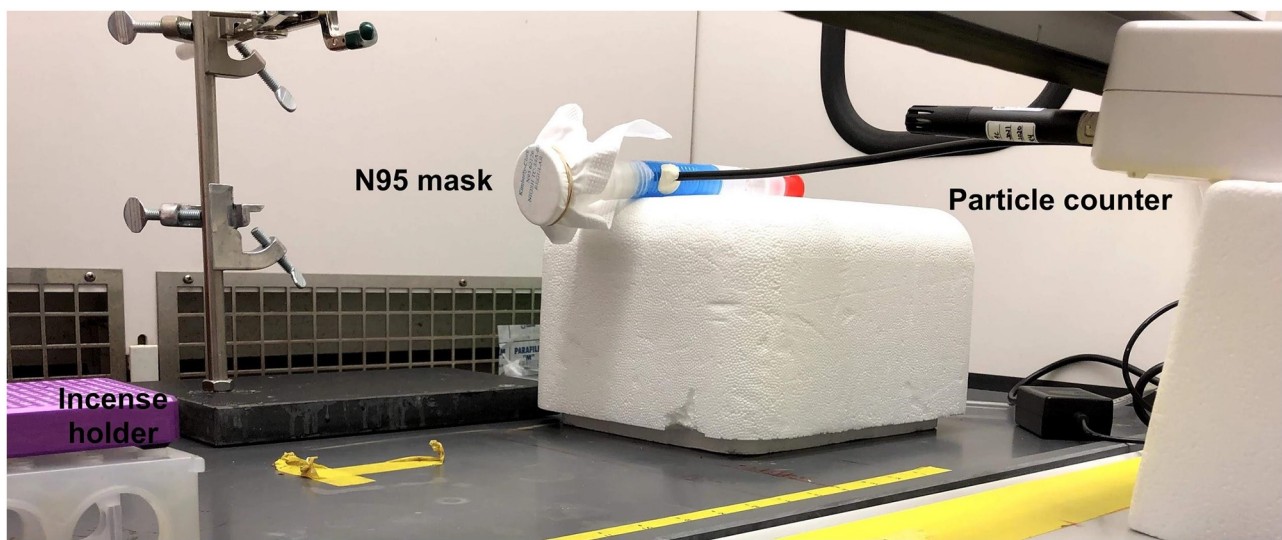

**Fig 3. Filtering efficiency test setup to test filter efficiency of a N95 mask using LightHouse handheld particle counter to detect particles of size 0.3 μm—10 μm produced by burning an incense stick.**

100 gram, and connectors (universal cuff adaptor, teleflex multi-adaptor). The detector measures particles at an airflow rate of 2.83 L/min, and reports particle counts for 4 different size bins: 0.3–0.49 µm, 0.5–0.99 µm, 1.0–2.49 µm, and 2.5–4.99 µm. The incense produces particles of various sizes, including those in the range picked up by the detector (0.3 µm—10 µm). We first measure the number of particles produced by the incense in each of 4 particle size bins. Then, we place the filter on the setup and run the particle counter to measure the number of unfiltered particles (note the incense is moved back the length of the filter mount away from the original position to maintain similar incense emission). To calculate the filtration efficiency for a given particle size range, we calculate the ratio of unfiltered particles in that range to the number of particles produced by the incense in that range, and then subtract from one. The filter efficiencies tested for Kimberly Clark N95 masks after repeated heating cycles, for particles sized 0.3–04.99 µm, are reported below in Fig 4. Filtration efficiency after 5 treatment cycles for particle sizes after 5 treatment cycles are reported in Table 1. Note the filter efficiency testing is done using a simple in house experimental setup as opposed to the standard testing which typically uses the TSI Automated filter tester 8130A.

Future work may involve carrying out a long term study to measure changes in filtration efficiency after treatment cycles following intermittent exposures. It has been shown that intermittent, low-level sodium chloride aerosol loading of N-95 FFR's, over periods of greater than 100 days, resulted in a degradation of filter efficiency, with the efficiency of some manufacturer models dropping below 95% [20]. However, this is not expected to be an issue within the limited 5 treatment cycles investigated in this study. Furthermore, we note from the referenced study that the long term usage was not was not accompanied by a significant increase in breathing resistance, indicating that long term usage does not result in a reduction in airflow [20].

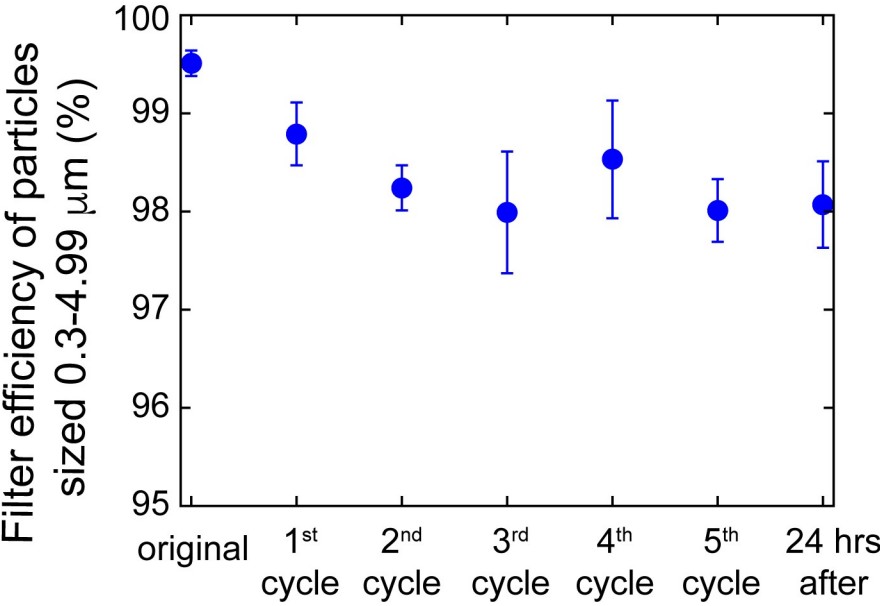

**Fig 4. Filtering efficiency of particles sized 0.3–4.99 µm, at an intake rate of 2.83 l/min, of Kimberly Clark N95 respirators undergoing cycles of heat treatments using the method detailed in this protocol with 2L of boiled water.**

Table 1. Filtering efficiency of particles for different particle sizes after the 5th treatment cycle.

| Filtration Efficiencies for Specific Particle Sizes after 5 treatments | |
|---|---|
| *Particle Size (microns)* | *Filtration Efficiency (%)* |
| *0.3–0.49* | *98 ± 0.4* |
| *0.5–0.99* | *98.2 ± 0.3* |
| *1.0–2.49* | *98.6 ± 2.5* |
| *2.5–4.99* | *100 ± 0.0* |

## Effect of changing water volume

It may be desirable to achieve time and temperature parameters greater than 65˚C for 30 minutes. A change in the dimensions of containers or water volumes used in the system will result in a change in the maximum temperature, and the time spent at that temperature. In particular, the cooling rate of the system is dependent upon the large vessel surface area to water volume ratio where the higher the surface area to water volume ratio, the higher the cooling rate. The effect of increasing boiled water volume to 3L while maintaining vessel sizing is shown in Fig 5a. This test was carried out without FFRs. The maximum achieved temperature increases along with time spent at a given temperature. However, as the vapour pressure is higher at those higher temperatures, it takes longer to reach the required relative humidity, and the humidity reached is lower (10 minutes to achieve 50% for 2L vs 17 minutes for 3L). This could be adjusted by increasing the size or number of soaked folded paper towels included in the FFR container.

The time spent simultaneously above a temperature threshold (65˚C or 70˚C) and the humidity threshold of 50% is plotted for 2L and 3L boiling volumes in Fig 5b. Use of 3L boiled water would allow for 70˚C moist heat to be applied for over 35 minutes, or 65˚C moist heat to be applied for almost 60 minutes.

## Filtration efficiency after treatment in Ziploc bags

It is desirable to consider how this method could be implemented using other materials based on local availability. A variant of this method was implemented using a smaller vessel (a 2.5L

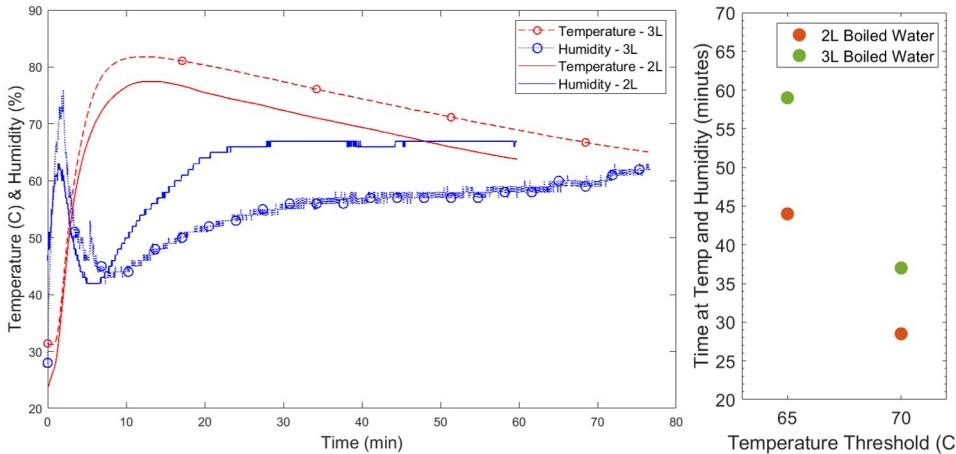

**Fig 5.** a. (left): Temperature and humidity inside the container over a single long cycle when placed in a vessel with 2L boiled water (solid lines) and 3L boiled water (dotted lines with circular markers). b (right): Time spent simultaneously above both humidity (50%) and temperature (65 & 70 ˚C) thresholds for 2L and 3L boiled water volumes.

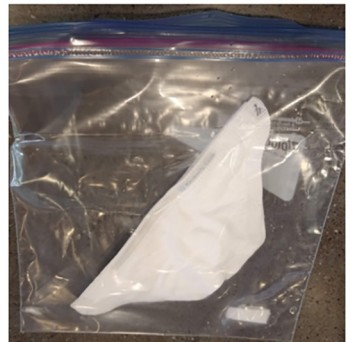
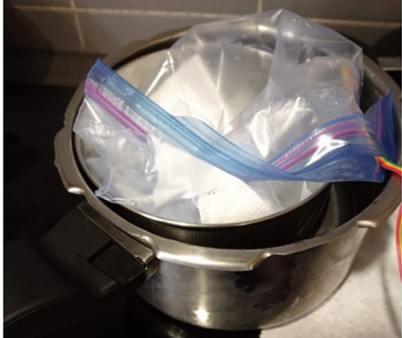

**Fig 6. (a) Kimberly Clark N95 respirator inside a Ziploc bag with a wet folded 5x5cm paper towel, (b) Ziploc bag with respirator resting on steel saucepan inside pressure cooker holding boiled water after being taken off stove.**

pressure cooker), with 1.5L water volume, where Ziploc bags were used to hold the respirators instead of the large Pyrex container. To avoid having the Ziploc directly contact the boiling water, they rested in an open steel saucepan floating in the water.

As previously discussed, a 5x5cm folded wet paper towel was placed inside the Ziploc bag along with the respirator halves (Fig 6a) and the sensor. Water in the open pressure cooker was brought to a boil and removed from heat. The Ziploc bag, resting in a steel saucepan, was immediately placed in the pressure cooker (Fig 6b), which was then sealed and allowed to sit for 45 minutes. It was noted during implementation of this procedure that it was challenging to reliably keep the moist towel and associated water droplets separate from the respirator, representing a risk of filtration piece getting wet, which could affect filtration efficiency. While temperature and humidity targets were reached, filtration efficiency of one of the respirator halves was measured to be below 95% after cycle 3 (Fig 7).

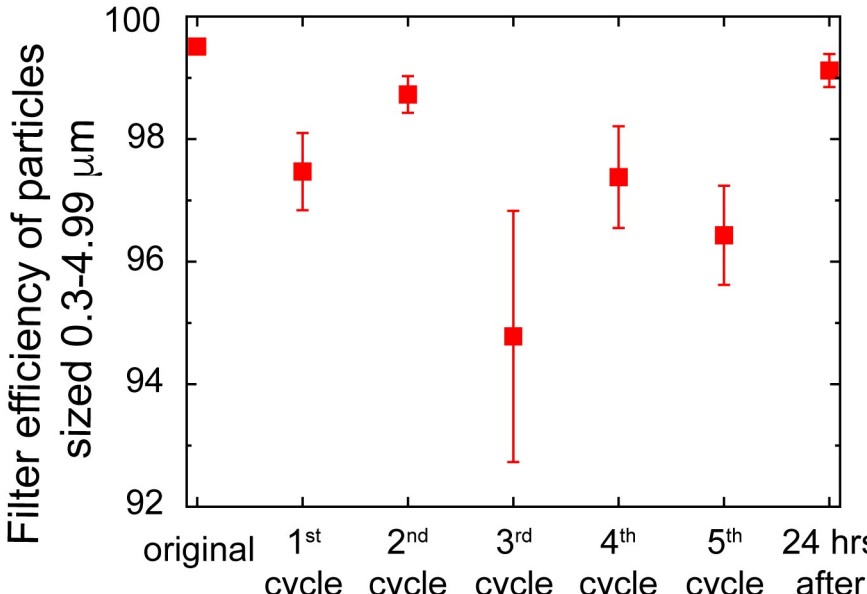

**Fig 7. Filtration efficiency of particles sized 0.3–4.99 μm at an intake rate of 2.83 L/min as outlined previously, undergoing cycles of heat treatments while placed in Ziploc bags.**

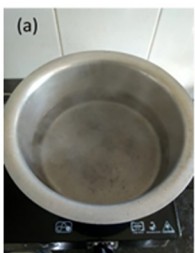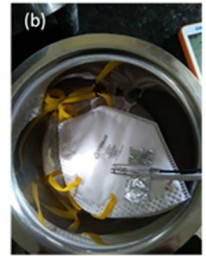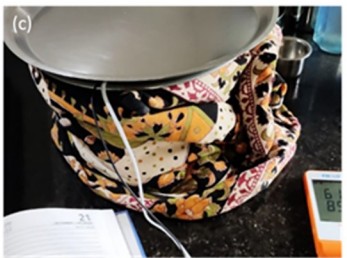

**Fig 8.** (a) Large aluminium 5L vessel with 2L of boiling water (b) smaller 2L steel vessel with Venus 4400 N95 FFR with temperature and humidity probes taped to the FFR. (c) Outer vessel covered with a plate, with a cloth wrapped around it for insulation to retain heat longer.

This indicates that holding N95 respirators in Ziploc bags while implementing this method may risk reducing filtration efficiency below 95%, though further study is required to gain confidence in this assessment.

## Implementation in India with materials relevant to South Asia

Most kitchens in India have steel or aluminium vessels typically with fitting metal lids. To verify that this method can be implemented in a variety of global settings, we implement the humid heat decontamination method using a 5L aluminum outer vessel with a 2L stainless steel inner container. We additionally wrap the outer vessel with cloth to retain heat longer. A Venus 4400 N95 respirator, one of the most common brands used in India, was chosen for the experiments. Five cycles were carried out using the configuration shown in Fig 8. Further experimental details are outlined in Supporting Information S1.1 in S1 File.

For the 6th cycle, to illustrate the scalability of the method, we used a 4-plate "idli cooker" where 4 respirators could be loaded, one at each level, as shown in Fig 9(a) and 9(b). As in the previous experiments the 4-plate vessel was kept in a closed container, hence not exposing the masks directly to steam. The use of such steamers may represent a route to greater scalability of the method, as large idli steamers, or other large dumpling steamers commonly found in South East Asia could be configured to hold up to 40–50 respirators (see Supporting Information S1.3 in S1 File). However, as the heat transfer characteristics of a larger system would be different, further work would be required to validate the method specifically for a larger system.

The temperature and humidity profiles in the different cycles are shown in Fig 10. In this implementation, measured temperature and humidity exceeded targets for a longer period than the initial implementation described in Figs 1 and 2. Temperature and humidity reached 65°C within 5–10 minutes and remained above 65°C for over 40 minutes even in the case without insulation. In most cases the temperature remained above the higher threshold of 70 °C for ~30 minutes in the un-insulated case, and ~45 minutes in the insulated case. This is expected to be partly due to the higher ambient temperature in Mumbai, and the effect of the insulation.

A simple home-built experimental setup for measuring filtration efficiency (see Supporting Information S1 in S1 File. 2 for further details) was used to evaluate the particulate filtration efficiency at 0.3 μm at a flow rate of 10 litres per minute for the India dataset. The data are shown in Fig 11. Filtration efficiency consistently remained above ~94% across all trials, showing no degradation in efficiency due to the heat treatment.

A similar experiment was carried out to explore the decontamination of surgical masks, the data are presented in the Supporting Information S1.4 in S1 File.

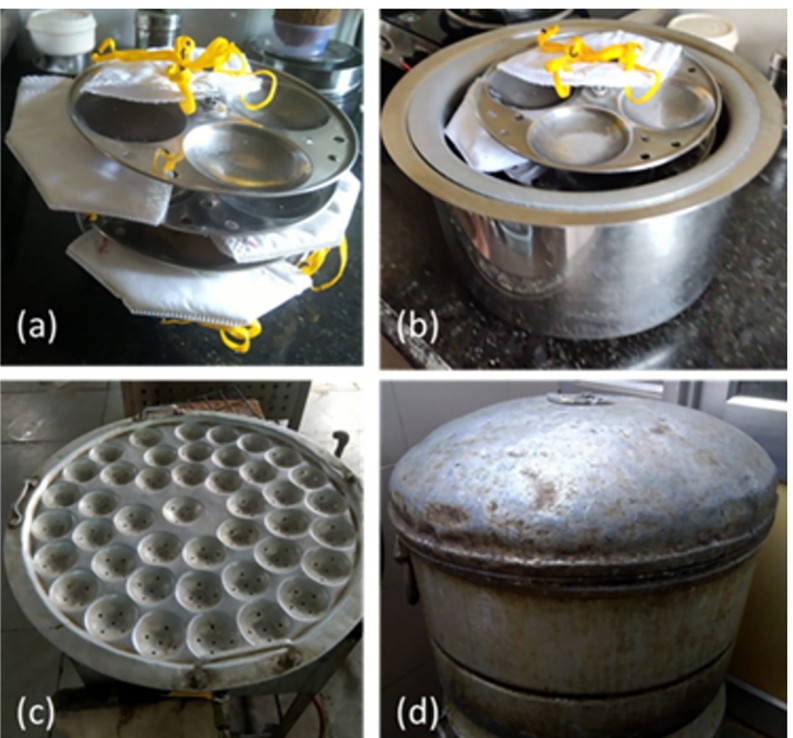

**Fig 9. Multiple respirator decontamination possibilities using the boiled hot water technique.** (a) 4-stand "idli-cooker" insert used for cycle 6, shown with 4 Venus 4400 respirators. (b) shows the nested vessels (without lids) used for this experiment.

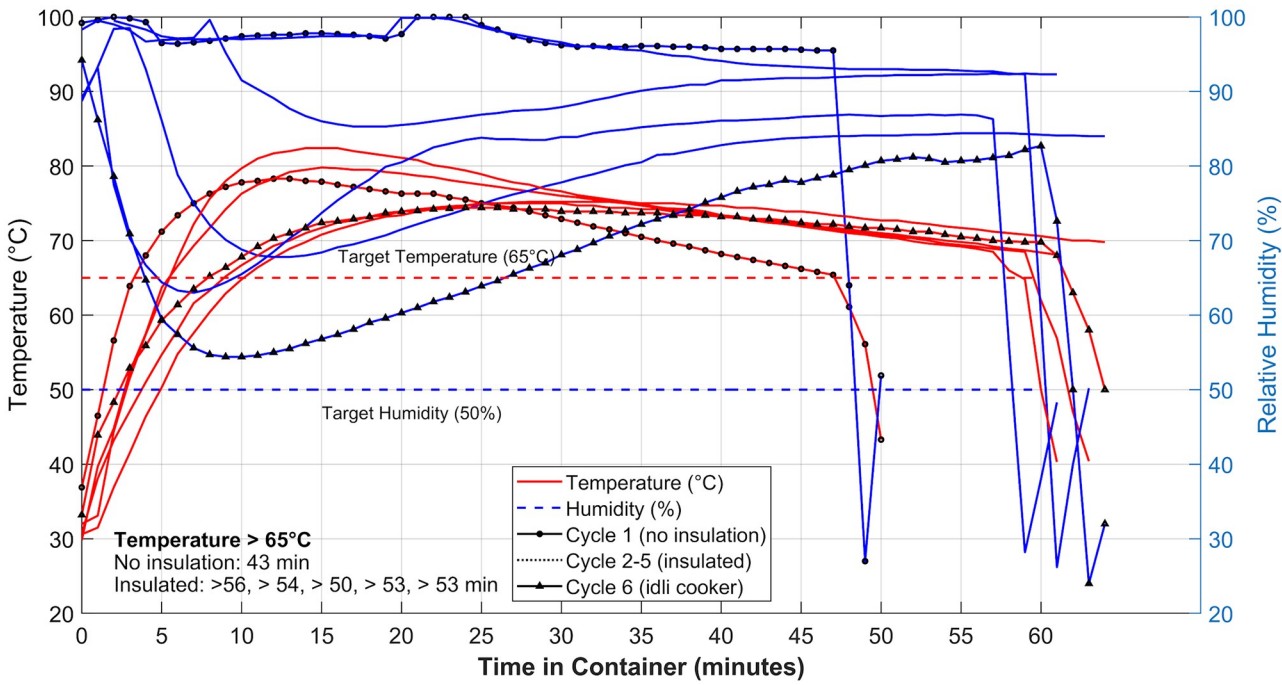

**Fig 10. Temperature and relative humidity inside the container holding the single N95 FFR during the 5 treatment cycles and the 6th cycle in an "idli-cooker" (container with 4 FFR capacity).** A sudden decrease in humidity followed by a recovery is noticed on opening the container at the end of the run. (For two cycles in which there was either no sealing or a leak in the seal of the inner vessel the RH values are close to saturation).

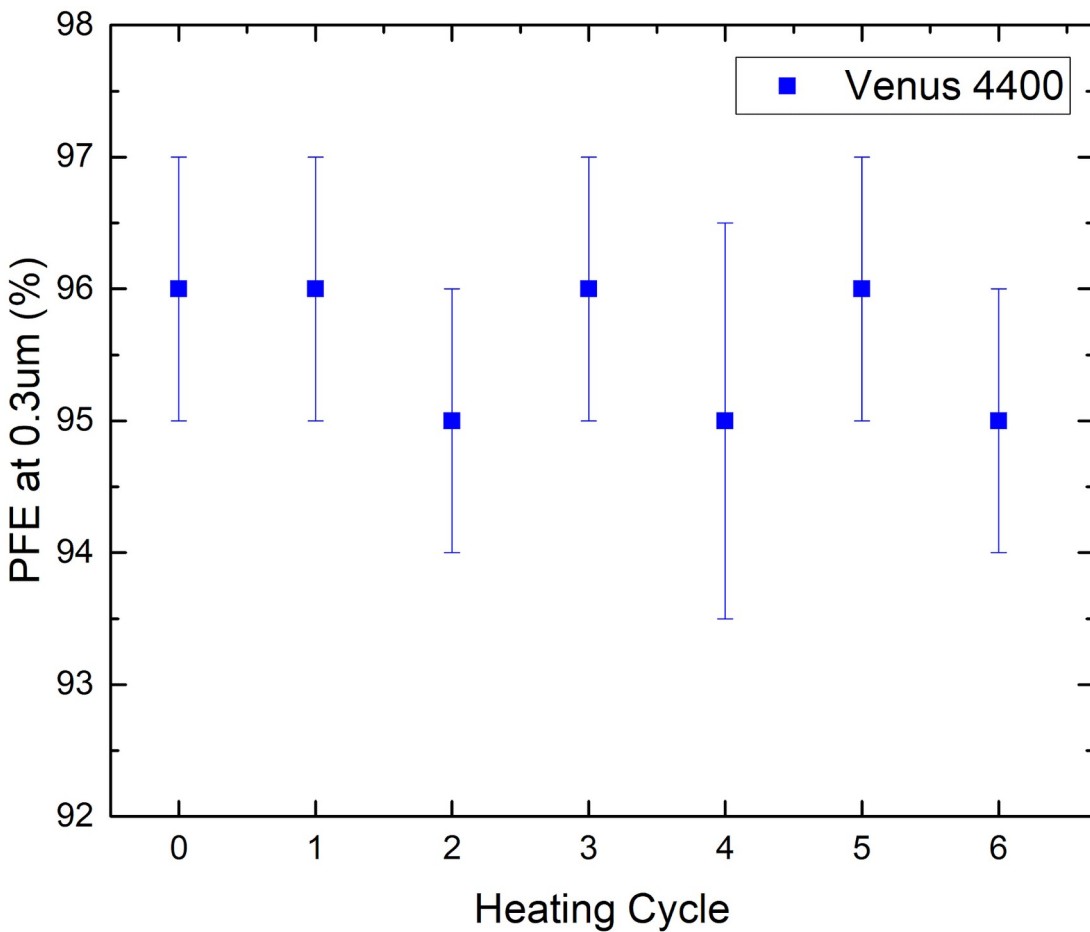

**Fig 11. Particulate filtration efficiency at 0.3 μm for the pristine FFR and after each decontamination and heating cycle.**

## Conclusions

Here we demonstrate a highly accessible heat-based protocol for N95 respirator decontamination that subjects them to >65°C and 50% humidity for over 30 minutes without requiring any advanced instrumentation or electricity. In this protocol, a 2L pot of water is brought to a rolling boil and removed from heat. Then, a Kimberly Clark N95 respirator and a damp paper towel are placed within a sealed container which is immersed in the boiled water for the treatment duration. The minimum and average filtration efficiencies measured after 5 treatment cycles of this method remained above 97% for particles sized 0.3–4.99 μm at a flow rate of 2.83 L/min, indicating filtration efficiency was not affected by this decontamination method. This outcome is consistent with previously reported literature, where many N95 FFR models have been shown to survive treatments of similar heat and humidity conditions without loss of filtration efficiency. Increasing the boiled water volume while maintaining vessel size allows for a longer treatment duration at high temperatures. A second implementation of this protocol was carried out on a Venus 4400 N95 FFR using standard sized utensils commonly found in South Asia. It achieved similar temperatures and humidities (>70°C at 50–90% humidity for over 30 minutes) and also maintained filtration efficiencies consistently above 94% for 6 treatment cycles. Higher treatment temperatures, such as 70°C for 30 minutes, or longer treatment durations can be achieved by increasing the boiled water volume while maintaining vessel size,

or by insulation of the vessel. We also show that the same method can be applied for the decontamination of surgical masks.

We reiterate that although these protocols were not tested on FFRs inoculated with SARS-CoV-2, they either use disinfection precedents set by either SARS-CoV or recent data based on inactivation of SARS-CoV-2 on FFR surfaces. While further work is still required to ensure viral inactivation of SARS-CoV-2 on FFRs under the given conditions, we believe this simple method can be easily modified to address any changes in guidelines for required heat and humidity.

We believe this simple method is robust to variations that may be encountered in individual settings, and provides a reliable, low-cost route for N95 decontamination applicable in resource-constrained settings anywhere in the world.

## Supporting information

**S1 File.**
(PDF)

## Acknowledgments

The methods used in this study were informed by insights from summary reports on heat-based decontamination of N95 masks put out by the N95 decon consortium (www. N95DECON.org). SK, UV and AB thank the Soft Matter Lab at TIFR and Emroj Hossain for help with the development of the PFE test setup.

## Author Contributions

**Conceptualization:** Siddharth Doshi, Arnab Bhattacharya, Manu Prakash.

**Data curation:** Siddharth Doshi, Samhita P. Banavar, Eliott Flaum, Surendra Kulkarni, Ulhas Vaidya, Shailabh Kumar, Tyler Chen, Arnab Bhattacharya.

**Formal analysis:** Siddharth Doshi, Samhita P. Banavar, Eliott Flaum, Surendra Kulkarni, Ulhas Vaidya, Shailabh Kumar, Arnab Bhattacharya.

**Funding acquisition:** Manu Prakash.

**Investigation:** Siddharth Doshi, Eliott Flaum, Surendra Kulkarni, Arnab Bhattacharya, Manu Prakash.

**Methodology:** Siddharth Doshi, Samhita P. Banavar, Eliott Flaum, Surendra Kulkarni, Arnab Bhattacharya, Manu Prakash.

**Project administration:** Siddharth Doshi, Manu Prakash.

**Resources:** Siddharth Doshi, Samhita P. Banavar, Eliott Flaum, Surendra Kulkarni, Ulhas Vaidya, Shailabh Kumar, Arnab Bhattacharya, Manu Prakash.

**Software:** Eliott Flaum, Arnab Bhattacharya.

**Supervision:** Arnab Bhattacharya.

**Validation:** Siddharth Doshi, Arnab Bhattacharya.

**Writing – original draft:** Siddharth Doshi, Arnab Bhattacharya.

**Writing – review & editing:** Siddharth Doshi, Samhita P. Banavar, Eliott Flaum, Surendra Kulkarni, Shailabh Kumar, Tyler Chen, Arnab Bhattacharya, Manu Prakash.

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
