## [Decision Letter · Decision Letter 0]

19 Nov 2020

PONE-D-20-33052

Applying Heat and Humidity using Stove Boiled Water for Decontamination of N95 Respirators in Low Resource Settings

PLOS ONE

Dear Dr. Prakash,

Thank you for submitting your manuscript to PLOS ONE. After careful consideration, we feel that it has merit but does not fully meet PLOS ONE’s publication criteria as it currently stands. One referee has suggested major and others minor and the manuscript needs to be revised based on all the referee comments. Therefore, we invite you to submit a revised version of the manuscript that addresses the points raised during the review process.

Please note that referees might have suggested references in their comments, however, is it upto the wish of authors to include relevant references in the manuscript.

Please submit your revised manuscript as soon as possible by Jan 02 2021 11:59PM. If you will need more time than this to complete your revisions, please reply to this message or contact the journal office at plosone@plos.org. Please include the following items when submitting your revised manuscript:

We look forward to receiving your revised manuscript.

Kind regards,

Prof. Dr. Yogendra Kumar Mishra

Academic Editor

PLOS ONE

Additional Editor Comments:

Looking forward to read your revised manuscript.

Journal Requirements:

Reviewers' comments:

Reviewer's Responses to Questions

**Comments to the Author**

1. Is the manuscript technically sound, and do the data support the conclusions?

Reviewer #1: No

Reviewer #2: Yes

Reviewer #3: Yes

2. Has the statistical analysis been performed appropriately and rigorously? 

Reviewer #1: I Don't Know

Reviewer #2: Yes

Reviewer #3: Yes

3. Have the authors made all data underlying the findings in their manuscript fully available?

Reviewer #1: Yes

Reviewer #2: Yes

Reviewer #3: Yes

4. Is the manuscript presented in an intelligible fashion and written in standard English?

Reviewer #1: Yes

Reviewer #2: Yes

Reviewer #3: Yes

5. Review Comments to the Author

Reviewer #1: Following are some of the suggestions.     

What is the level of COVID 19 virus contamination of the used membrane used in this study? Also, provide the %decontamination of the COVID19 virus with your method?

Does recycle reduce the airflow? Please comment on it. Reduction in airflow may impact the breathing capacity. Generally speaking, the membrane capacity to capture viruses or particles reduces with longer usage. It is due to pore blockage by particles or viruses. The paper does not have any information about it.

The authors did not provide any pieces of evidence about why65C temperature and humidity more than 50% can kill the COVID19 virus. Please provide it in the results and the discussion section.

The authors have used Kimberly Clark N95membrane. The south Asian countries have different brand membrane than Kimberly Cark N95. The different brands may have different types of polymer material than Kimberly Clark. The different polymer may interact with humidity and temperature differently. Please comment on how your method will be useful in that scenario. Also, please provide details about the membrane material used in Kimberly Clark N95. Also, comment on how the material interacts with humidity and temperature. Most of our polyolefin type of material and heat/humidity may cause physical aging or plasticization which will change the porosity of the membrane.

Reviewer #2: The manuscript entitled “Applying Heat and Humidity using Stove Boiled Water for Decontamination of N95 Respirators in Low Resource Settings” submitted by Siddharth Doshi1 et al. is a nice and timely submission on current going pandemic scenario. The authors have excellently explained a novel method for decontamination process of N95 mask for their reuse. The data presented are excellent and reliable. Only some small typos and grammar mistakes has to be taken care off. It can be recommended for acceptance after some minor revision as commented below:

Units like 0.3-4.99um should be written as 0.3-4.99 um.

Line 41: change to 650C. Please check all over the manuscript.

Some recent citation are suggested to be included like: DOI: 10.1126/sciadv.abb8097, https://doi.org/10.1080/14760584.2020.1811091, https://doi.org/10.1080/07391102.2020.1802348, https://doi.org/10.3389/fnano.2020.571284

Reviewer #3: Proposed research explores the effects of heat and humidity to decontaminate of N-95 respirators. This is a very affordable and scalalble approach, and very much needed to low recourse setting.

All studies in this article are well-planned and scientifically explained.

Keeping outcomes and prospects into consideration, I do recommend this work for publication.

Comment:

1. Critical and comparative analysis of state-of-art techniques

2. Challenges associated with proposed research

3. Can nano play any role in this techniques?

4. to cover wider aspects, better to discuss the aspects of diagnosis and therapy. Below citations can be useful

https://www.frontiersin.org/articles/10.3389/fnano.2020.571284/full

https://www.sciencedirect.com/science/article/pii/S2468519420300665

https://pubs.acs.org/doi/abs/10.1021/acsabm.0c01004

https://arxiv.org/abs/2006.08536

6. PLOS authors have the option to publish the peer review history of their article (what does this mean?). If published, this will include your full peer review and any attached files.

Reviewer #1:

Reviewer #2:

Reviewer #3:

---

## [Author Response · Author response to Decision Letter 0]

13 Jul 2021

Rebuttal Letter 

Dear Editor: 

We are submitting our response to comments raised by academic editors to the manuscript titled “Applying Heat and Humidity using Stove Boiled Water for Decontamination of N95 Respirators in Low Resource Settings”. We provide a response to the journal comments and specific reviewer comments below, itemizing our responses by reviewer. Original reviewer comments are highlighted in blue. We periodically will refer to references by number, to avoid confusion, these refer to the respective references in the main manuscript file. 

Journal Requirements:

1. Please ensure that your manuscript meets PLOS ONE's style requirements, including those for file naming. The PLOS ONE style templates can be found at:

We acknowledge the editor’s request to ensure we comply with PLOS ONE formatting requirements and have accordingly made changes, including to the case of the article title. To the best of our knowledge we believe the revised manuscript is compliant with PLOS ONE style requirements.

Reviewer #1: 

We thank the reviewer for their careful reading of our manuscript. Before addressing their comments on a point-by-point basis, we wish to make a general clarification on the scope of our paper. The contribution of this paper is not on reporting and validating a set of novel treatment conditions for achieving viral inactivation of SARS-CoV-2 on membranes. We emphasise in the manuscript that we make no attempt to identify what temperature and humidity conditions actually cause SARS-CoV-2 inactivation. Rather, we parse prior literature on viral inactivation of FFR respirators by heat and humidity. The set of treatment conditions we use in our work represent current scientific consensus for achieving decontamination while maintaining FFR integrity. These have previously been implemented using equipment available in high resource settings in published literature, including the work of Anderegg et al., (PLOS ONE, July 2020), and the work of Liao et al., (ACS Nano, May 2020), demonstrating that heat with humidity is a promising approach. 

Furthermore, we don’t try and investigate broad questions about the use of FFRs and degradation of their efficacy over long periods of time in general settings. These questions are explored in literature that we cite and are also deserving of separate detailed studies. Rather, we focus our efforts on investigation of a novel decontamination protocol with specific applicability to low resource settings, and verification of FFR efficacy for a limited number of treatment cycles. 

The contribution of this paper lies in developing a method to achieve previously established treatment conditions using simple equipment that is easily applicable in low-resource settings, and demonstrating that this specific implementation does not degrade filtration efficiency in different settings (California, Mumbai). Furthermore, it is to our knowledge, the first scientifically-documented heat-based method to allow for decontamination without electricity, requiring only a heat source, water and common kitchen utensils. This is likely to be a valuable reference for the field of low-resource decontamination. 

• What is the level of COVID 19 virus contamination of the used membrane used in this study? Also, provide the %decontamination of the COVID19 virus with your method?

We do not use membranes inoculated with SARS-CoV-2 in this study.

This work does not study viral inactivation of the SARS-CoV-2 virus. SARS-CoV-2 viral samples are not readily accessible, and a systematic study of viral inactivation is outside the scope of this paper. As outlined in the general response to Reviewer 1, this paper begins by evaluating literature on viral inactivation by application of heat and humidity, from which we arrive at the set of parameters used in this study. This evaluation is found in our introduction and references sources [1-9]. We discuss this further in our response to the fourth point raised by Reviewer 1. The contribution of this paper is on providing a method of achieving this set of heat and humidity conditions, which from current literature appear promising for disinfection, in low resource settings. 

• Does recycle reduce the airflow? Please comment on it. Reduction in airflow may impact the breathing capacity. 

We thank Reviewer 1 for raising this comment. 

We address this point in our updated manuscript, where we include a reference [20] and discussion of this point. We note from the referenced study, where N95 FFR’s were loaded with intermittent, low-level sodium chloride aerosol over periods of greater than 100 days, that long term usage was not was not accompanied by a significant increase in breathing resistance. This indicates that long term usage does not result in a reduction in airflow [20]. We do not explicitly measure airflow in our study to assess the impact of decontamination. However, based on this evidence, we believe that for the limited number of re-uses that are being assessed, there is no concern around reduction in airflow. 

• Generally speaking, the membrane capacity to capture viruses or particles reduces with longer usage. It is due to pore blockage by particles or viruses. The paper does not have any information about it.

We thank Reviewer 1 for raising this comment. Firstly, we show within our manuscript that 5 aerosol loading and subsequent heat/humidity based treatment cycles do not adversely affect the filtration efficiency of the FFRs. Hence the membrane capacity to capture particles is not reduced. Based on the reviewer feedback, we discuss the issue of efficacy over time further in our updated manuscript, where we include a reference [20] alongside comments on this issue. 

As the reviewer states, the filtration efficiency of a membrane may decrease over time with greater usage, as observed in reference [20], where N95 FFR’s were loaded with intermittent, low-level sodium chloride aerosol over periods of greater than 100 days. This is expected to occur during usage of a respirator over a long period of time. This is an important point for industrial settings , where multi-use respirators may be used for a long period of time with continuous exposure to high particle loadings. 

However, this is not a cause for concern for N95 FFRs in the context that we investigate, where we study decontamination for a reasonable number of re-uses (up to 5 cycles). Pore size for N95 FFRs is large compared to the size of particles being captured in aerosol range. Significantly extended usage periods may have other issues such as material degradation over time or degradation of fit which put an upper bound on their usage time. These issues are investigated in references [13-16] and discussed in our introduction. While we do not attempt to establish the upper bounds of usage time, it is widely acknowledged [14-16] that 5 re-use cycles within the temperature and humidity ranges we investigate do not cause degradation of FFR efficacy, and shown once again experimentally in our work for our specific protocol.

• The authors did not provide any pieces of evidence about why65C temperature and humidity more than 50% can kill the COVID19 virus. Please provide it in the results and the discussion section.

We evaluate the literature in our introduction, from which we arrive at the parameters of 65°C and 50% humidity. These represent scientific consensus around parameters to achieve viral inactivation without degrading FFR integrity over a limited number of cycles. 

To outline the evidence we evaluate in the introduction:

“There is recent data suggesting that SARS-CoV-2 in liquid media can be inactivated by exposure to 70°C for 5 minutes [2], it appears that the virus requires a much longer heat-treatment for inactivation on N95 FFR surfaces. Recent reports indicate that application of 70°C dry heat for 30 min only led to a 1.9-log reduction in viral load of SARS-CoV-2 in an unknown media on N95 fabric [3], which is below the minimum 3-log inactivation level suggested by the FDA for emergency use authorization for decontamination [4]. Additionally, the media used for inoculation may not be representative of human saliva or mucin, which may falsely increase inactivation rates compared to a real-world scenario [5-8]. While direct studies involving the effects of heat and humidity on the SARS-CoV-2 virus on N95 FFRs are limited, increased humidity was found to increase inactivation rates of the H1N1 influenza virus at 65C on steel surfaces [10-12]. A non-peer-reviewed report found that increasing relative humidity from 13% to 48% increased inactivation of MS2 and Phi6 bacteriophages deposited with PBS on N95 FFRs by >4-log during a 72oC heat treatment for 30 min [7]. After this 30 min treatment at 72oC and 48% relative humidity, both viruses were inactivated by over 6-log. MS2 belongs to a class of non-enveloped viruses, which are traditionally considered to be more difficult to inactivate than enveloped viruses such as SARS-CoV-2 [9]. Given this emerging data, it is likely, though unproven, that temperatures of 65oC and relative humidities of >50% applied for at least 30 minutes will lead to at least 3-log inactivation of SARS-CoV-2 on an N95 FFR.”

This evaluation was present in the introduction of the original manuscript. For clarity, we include another brief mention of this in the conclusion section, where we outline that our treatment conditions were set by disinfection precedents set by either SARS-CoV or recent data based on inactivation of SARS-CoV-2 on FFR surfaces. We believe the full flow of our argument and discussion of a variety of sources from literature is best established in the introduction. It would be detrimental to include a partial or incomplete discussion elsewhere. 

• The authors have used Kimberly Clark N95membrane. The south Asian countries have different brand membrane than Kimberly Cark N95. The different brands may have different types of polymer material than Kimberly Clark. The different polymer may interact with humidity and temperature differently. Please comment on how your method will be useful in that scenario. Also, please provide details about the membrane material used in Kimberly Clark N95. Also, comment on how the material interacts with humidity and temperature. Most of our polyolefin type of material and heat/humidity may cause physical aging or plasticization which will change the porosity of the membrane.

We thank Reviewer 1 for raising this comment. Firstly, we implemented our original experiments within our manuscript with two different models of N95 FFR. Experiments in California were conducted with a Kimberly Clark N95 FFR. Experiments in Mumbai were conducted with a Venus 4400 N95 FFR. Both maintained their filtration efficiencies over the treatment duration. To emphasise this point further based on reviewer feedback, we explicitly name the two different brands in the conclusion to ensure clarity. Most N95 (or equivalent e.g. FFP2 standard) mask vendors across the world use melt-blown polypropylene as the central filtration layer, and while we report results for two commonly used brands, we believe this would be universally applicable to all N95 or equivalent masks.

Secondly, to take into account this feedback, we also carry out 5 cycles of treatment and filtration efficiency measurement on surgical masks and include this result in the Supporting Information. This extends our work on heat and humidity treatment to another mask type which is becoming increasingly important for protection of the general public. 

Finally, in the introduction we discuss reports in the literature [13-16] where heat and humidity were applied to many models of N95 FFR. A microscopically precise understanding of the impact of heat and humidity on the material is currently not complete. As such, it is difficult to comment precisely on such effects. As the reviewer states, there are risks of material degradation or changes in membrane properties through processes such as plasticisation. Therefore a balance of temperature, humidity, and duration of decontamination must be carefully chosen to strike a balance between viral inactivation and N95 performance [13]. Currently, the best evaluation of the impacts of humidity and temperature treatments is by empirical demonstration. Fortunately, many models of N95 FFR have been shown to undergo at least three cycles of elevated temperature (65–85C) at greater than 50% relative humidity for 20–30 min while maintaining filtration efficacy and fit [14, 15, 16]. This indicated that conditions of moist heat at 65-80C, with 50% relative humidity for 30 minutes may be a promising target for decontamination of SARS-CoV-2 on N95 FFRs, leading to our choice of decontamination parameters.

We believe our method, for 5 cycles, would be universially applicable to most N95 masks. 

 

Reviewer #2: 

• The manuscript entitled “Applying Heat and Humidity using Stove Boiled Water for Decontamination of N95 Respirators in Low Resource Settings” submitted by Siddharth Doshi1 et al. is a nice and timely submission on current going pandemic scenario. The authors have excellently explained a novel method for decontamination process of N95 mask for their reuse. The data presented are excellent and reliable. Only some small typos and grammar mistakes has to be taken care off. It can be recommended for acceptance after some minor revision as commented below:

• Units like 0.3-4.99um should be written as 0.3-4.99 um.

• Line 41: change to 650C. Please check all over the manuscript.

We thank the reviewer for their helpful comments. We have corrected the typographical errors the reviewer refers to and have carefully checked the rest of the document as suggested. 

• Some recent citation are suggested to be included like: DOI: 10.1126/sciadv.abb8097, https://doi.org/10.1080/14760584.2020.1811091, https://doi.org/10.1080/07391102.2020.1802348, https://doi.org/10.3389/fnano.2020.571284

Regarding the suggested citations, we note that they are on the general areas of bionanotechnology and immune informatics, or very specific works on multi-epitope vaccine design, and structural biology of the receptor binding domain. We thank the reviewer for this interesting set of reading. However, it is unclear to us how this provides relevant background to work on N95 FFR decontamination in low resource settings. Therefore, to ensure that the paper remains tightly focused, we are not including the recommended citations. 

 

Reviewer #3: 

Proposed research explores the effects of heat and humidity to decontaminate of N-95 respirators. This is a very affordable and scalalble approach, and very much needed to low recourse setting.

All studies in this article are well-planned and scientifically explained.

Keeping outcomes and prospects into consideration, I do recommend this work for publication.

Comment:

1. Critical and comparative analysis of state-of-art techniques

2. Challenges associated with proposed research

3. Can nano play any role in this techniques?

4. to cover wider aspects, better to discuss the aspects of diagnosis and therapy. Below citations can be useful

https://www.frontiersin.org/articles/10.3389/fnano.2020.571284/full

https://www.sciencedirect.com/science/article/pii/S2468519420300665

https://pubs.acs.org/doi/abs/10.1021/acsabm.0c01004

https://arxiv.org/abs/2006.08536

We thank the reviewer for their helpful comments. We discuss comparable techniques in our introduction, where we evaluate references [3-16] along with challenges in decontamination of N95 FFR’s. We wish to ensure the paper clearly communicates our results on N95 FFR decontamination for low resource settings. In order to keep the focus on low resource implementations of a specific technique, we will avoid discussing broad areas such as “nano”, or diagnosis and therapy, which are widely covered in the literature. 

Regarding the suggested citations, we note that they are on the general areas of electrochemical sensors, nanosensors and bionanotechnology. We thank the reviewer for this interesting set of reading. However, it is unclear to us how this provides relevant background to work on N95 FFR decontamination in low resource settings. Therefore, to ensure that the paper remains tightly focused, we are not including the recommended citations.

---

## [Editor Report · Decision Letter 1]

15 Jul 2021

Applying Heat and Humidity using Stove Boiled Water for Decontamination of N95 Respirators in Low Resource Settings

PONE-D-20-33052R1

Dear Dr. Prakash,

We’re pleased to inform you that your manuscript has been judged scientifically suitable for publication and will be formally accepted for publication once it meets all outstanding technical requirements.

Kind regards,

Yogendra Kumar Mishra, Ph. D.

Academic Editor

PLOS ONE

Additional Editor Comments (optional):

Authors have carefully addressed the remarks suggested by the referees and the revised paper is in very good stage and is ready for publication. Acceptance is recommended.
---

## [Editor Report · Acceptance letter]

23 Sep 2021

PONE-D-20-33052R1 

Applying Heat and Humidity using Stove Boiled Water for Decontamination of N95 Respirators in Low Resource Settings 

Dear Dr. Prakash:

I'm pleased to inform you that your manuscript has been deemed suitable for publication in PLOS ONE. Congratulations! Your manuscript is now with our production department. 

Kind regards, 

on behalf of

Professor Yogendra Kumar Mishra 

Academic Editor

PLOS ONE